# OpenReview forum: "The Alignment Waltz: Jointly Training Agents to Collaborate for Safety"
_ICLR.cc/2026/Conference — ICLR 2026 Poster_

### Official Review · Reviewer_gQyF · 2025-10-27

**Soundness:** 3
**Presentation:** 4
**Contribution:** 3
**Rating:** 6
**Confidence:** 4

**Summary:**

This paper addresses the critical challenge of balancing helpfulness and harmlessness in LLMs, aiming to mitigate both adversarial vulnerability and overrefusal on benign prompts. The authors propose WALTZRL, a novel multi-agent reinforcement learning framework that formulates safety alignment as a collaborative, positive-sum game. This system jointly trains a conversation agent and a feedback agent, where the latter is incentivized by a Dynamic Improvement Reward to provide useful suggestions that improve the conversation agent's response, rather than simply blocking it. At inference, this feedback mechanism is applied adaptively, preserving low latency on safe queries. The paper's primary contribution is this collaborative framework, which, as demonstrated across five datasets, significantly reduces both unsafe responses (e.g., from 39.0% to 4.6% on WildJailbreak) and overrefusals (from 45.3% to 9.9% on OR-Bench), effectively advancing the Pareto front between safety and utility without degrading general capabilities.

**Strengths:**

1. Novelty of Method: The paper proposes a novel multi-agent RL method (WALTZRL) that frames the alignment problem as a collaborative, positive-sum game. This is a significant departure from standard safeguard models, which act as blunt classifiers, or other multi-agent approaches that often rely on zero-sum (adversarial) setups. By jointly training a conversation agent and a feedback agent to cooperate, this framework creates a new mechanism for adaptive, fine-grained safety corrections that can improve a response rather than just rejecting it.

2. Insightful Analysis: The authors provide insightful analysis and thorough ablation studies that successfully validate their core design choices. The investigation into the DIR variants in Section 3.3 is particularly strong, clearly demonstrating why conditioning the improvement reward on label correctness is crucial for balancing feedback usefulness and label accuracy. Furthermore, the justification for the two-stage training process effectively shows how it overcomes the sample diversity challenge as the conversation agent improves.

3. Clarity and Presentation: The paper is exceptionally well-written, and the overall presentation is clear, logical, and easy to follow. The "Alignment Waltz" metaphor is effective, and figures like Figure 1 provide an excellent, intuitive overview of both the inference-time protocol and the RL training step. This clarity makes the paper's novel contributions highly accessible despite the complexity of the underlying framework.

**Weaknesses:**

1. Limited Model Diversity: The experiments are confined to a single model architecture, Llama-3.1-8B, which is insufficient to claim broad generalizability. It is unclear how WALTZRL would perform with other prominent models, such as Qwen, or models known for different reasoning behaviors. Exploring the framework's effectiveness on models with diverse architectures and reasoning capabilities would significantly strengthen the paper's claims.

2. Simplified Datasets: The study relies on single-turn adversarial (WildJailbreak) and overrefusal (OR-Bench) datasets. These, while useful, do not reflect the complexity of multi-step, real-world interactions, such as those found in coding or computer-use agent datasets. The paper lacks analysis of how WALTZRL would handle more complex, emergent safety failures that can occur over extended, stateful interactions in agentic tasks.

**Questions:**

1. Can you provide additional results of Qwen?

---

> ### Author Response · Authors · 2025-11-20
> **Response to reviewer gQyF**
>
> We sincerely appreciate reviewer `gQyF` for your thoughtful comments. We appreciate that you find our WaltzRL method novel, our analysis insightful, and our paper “exceptionally well-written.” We now provide a few clarifications:
>
> > Limited Model Diversity
> We appreciate your suggestion to evaluate on additional backbones. WaltzRL's multi-agent RL with multi-turn rollouts makes each training run computationally expensive, so we focused our experiments on Llama-3.1-8B given our compute constraints. However, we are actively seeking additional resources to conduct experiments on other model families and will update this thread if we have new results!
>
> We would also like to emphasize that **our framework is conceptually model-agnostic, and our approach does not involve any model-specific tricks**. We therefore expect WaltzRL to transfer well to other model families and sizes without modification. We will include this discussion in our manuscript, and, if accepted, we plan to try our best to add results on additional models in the camera-ready version. Thanks again for your suggestion.
>
> > Simplified Datasets
>
> Thanks a lot for your suggestion on multi-turn datasets. To clarify, while WaltzRL involves multi-step interactions between the conversation and feedback agents, the scope of this work is focused on addressing single-turn user queries, a fundamental and widely used setting to study robustness against adversarial attacks and overrefusal.
>
> Since our goal is focused on building a general recipe that can enhance helpfulness and harmlessness simultaneously, we focus on single-turn queries to maintain a clear scope. We believe applying it to a specific real-world agentic setting, such as coding or computer-use agents, is a promising venue for future work. **We envision that our comprehensive study on single-turn queries can provide insights for extending the WaltzRL approach to multi-turn, stateful interactions.** That being said, if you have any specific suggestions on what additional datasets to include, we are happy to conduct further experiments on those settings!
>
> ---
>
> We look forward to addressing any remaining questions during the rebuttal and hope these updates warrant a higher evaluation.

---

> > ### Comment · Reviewer_gQyF · 2025-11-25
> > **Response to the reviewer's rebuttal**
> >
> > I thank the authors for their response, which has largely resolved my concerns. This paper presents a novel training paradigm for balancing LLM safety and helpfulness that proves effective across various datasets. I look forward to seeing future work extend these self-play and self-evolving methods to more complex tasks, such as real-world agentic applications, and seeing how they can further boost potential when combined with traditional alignment techniques. I view this paper as a pivotal contribution to the field and recommend a Strong Accept.

---

### Official Review · Reviewer_uDhH · 2025-10-29

**Soundness:** 3
**Presentation:** 3
**Contribution:** 3
**Rating:** 4
**Confidence:** 3

**Summary:**

This paper proposes a multi-agent reinforcement learning framework that formulates safety alignment as a collaborative, positive-sum game. A conversation agent generates responses to users’ prompts and a feedback agent provide revision feedback for the conversation agent to improve the responses. Both agents are optimized simultaneously using reinforcement learning with rewards obtained from LLM judgers. A dynamic improvement reward (DIR) is integrated to encourage the feedback agent to focus on the improvements made by the conversation agent.

**Strengths:**

The paper is overall easy to follow.
The idea is novel, compared to previous multi-agent framework where a zero-sum game is played between the agents, the proposed framework encourage the collaboration between agents, by designing the dynamic improvement reward that uses the improvements of conversation agent to guide the feedback agent.
The effectiveness of inference-time collaboration and RL-based joint training is well demonstrated.
Overall, the main contributions include: introduction of a multi-agent RL framework that optimizes two agents with a positive-sum game, introduction of the dynamic improvement reward that encourage the collaboration between the agents, and demonstration of the effectiveness of the proposed framework on multiple metrics.

**Weaknesses:**

The motivation is not clearly described, why the collaborative training can enhance helpfulness and reduce harmlessness?

The core drawback is the experimental setting where the collaborative round number is set to 1, I wonder whether such a setting is sufficient to fully harness/demonstrate the power of the proposed DIR. Also, it remains unknown whether the feedbacks are inherently consistent across collaborative rounds. There is a lack of ablation study to compare the performance of systems with and without DIR, currently, in Sec. 3.3, all (A), (B), and (C) settings include DIR. The stoping criteria is also not verified in the experiments.

The experiments should be also benchmarked with other multi-agent RL frameworks, e.g., (Zheng et al. 2024)

Zheng et al., Toward Optimal LLM Alignments Using Two-Player Games, 2024.

**Questions:**

The motivation may be improved, by describing the role that the feedback agent plays during training and inference, i.e., to provide reflection on over refusal for revision.

The ablation study on DIR, mentioned above, may be provided to better demonstrate its importance. The visualization of DIR can be also added to show the convergence of DIR.

There are grammatical errors that need to be fixed.

---

> ### Author Response · Authors · 2025-11-20
> **Response to reviewer uDhH (1/2)**
>
> We greatly appreciate reviewer `uDhH` for your detailed review. We are excited that you find our idea novel and the effectiveness of our approach is well demonstrated with multiple metrics. As a reminder, our main contributions are proposing the WaltzRL framework to **jointly optimize two agents that co-evolves for safety**, the **novel DIR reward formulation**, and experiments demonstrating that WaltzRL **simultaneously reduces both unsafe responses and overrefusals against strong baselines**. We now address your concerns:
>
> > The motivation is not clearly described, why the collaborative training can enhance helpfulness and reduce harmlessness?
>
> Our motivation for collaborative training is to address a key limitation of existing safeguard paradigms: because safeguards are trained separately from the LLM conversation agent, they can only block entire responses without providing feedback **that is actionable for the specific conversation agent**, thus they reduce unsafe content at the cost of overrefusal (detailed in L50).
>
> WaltzRL's multi-agent collaborative training addresses this through the DIR reward, which specifically incentivizes feedback that improves the conversation agent's performance, fostering deeper collaboration between the two agents. Because the feedback agent can **adaptively address either overrefusal or unsafe content via natural language feedback**, this approach enhances both helpfulness and harmlessness simultaneously. We will strengthen this motivation in Sections 1 and 2.2 of the revision, and are happy to provide further clarification if needed.
>
>
> > The core drawback is the experimental setting where the collaborative round number is set to 1
>
> Thank you for raising this point. To clarify, **the goal of WaltzRL is to produce a safe and non-overrefusing response as quickly as possible** so that latency and token costs are minimized, instead of achieving this through a fixed number of feedback rounds. Therefore, fewer feedback rounds are always preferable, as they indicate more efficient convergence to the desired outcome.
>
> We set $T_{max}$ to 1 because: (1) As shown in Section 3, $T_{max}=1$ already performs extraordinarily well, outperforming strong baselines and achieving single-digits ASR and ORR. Thus we believe $T_{max}=1$ is sufficient in our current setting. (2) Additional interaction rounds increase inference cost at deployment. With adaptive feedback $T_{max}=1$, our inference cost is at most comparable to applying a safeguard model plus generating a revised response. In summary, **we experiment with $T_{max}=1$ because it’s both practically desirable and already empirically strong**. We will include this discussion in our draft for further context.
>
> > There is a lack of ablation study to compare the performance of systems with and without DIR, currently, in Sec. 3.3, all (A), (B), and (C) settings include DIR
>
> **TL;DR: we tested ablating the DIR reward and found notably worse performance, demonstrating DIR’s effectiveness.**
>
> Following your insightful suggestion, we have conducted additional ablation studies by comparing the performance of WaltzRL with and without DIR. Specifically, for the variant without DIR, we use the final conversation agent response reward as the reward for the feedback agent. We show the training dynamics and results compared to WaltzRL below:
>
> Stage 1 training dynamics: [link](https://imgur.com/a/BIKtxWg)
>
> Stage 2 training dynamics: [link](https://imgur.com/a/iiOSLkP)
>
>
> | Method | WJ⬇️ | FH⬇️ | SR⬇️ | OB⬇️ | FB⬇️ | Avg ASR⬇️ | Avg ORR⬇️ | F1⬆️ |
> |---|---|---|---|---|---|---|---|---|
> | **Ours: WaltzRL** | **4.6** | **6.2** | **0.3** | 9.9 | **5.4** | **3.7** | 7.6 | **94.3** |
> | No DIR reward | 11.5 | 13.2 | 0.6 | **6.9** | 6.6 | 8.4 | **6.7** | 92.4 |
>
>
> As shown in the results above, ablating the DIR reward leads to lower %improved and notably worse outcome reward and final eval results across five datasets. We hypothesize that **DIR densifies the reward signal for the feedback agent by isolating the effect of feedback from the quality of initial conversation agent response**, thus stabilizing training. Without DIR, reward on the feedback is too coarse, as it becomes unclear whether the feedback itself enabled better response or whether the initial response was already strong.

---

> > ### Author Response · Authors · 2025-11-20
> > **Response to reviewer uDhH (2/2)**
> >
> > > The stopping criteria is also not verified in the experiments.
> >
> > We’d like to point out that Table 3 discusses the feedback agent label accuracy and the feedback triggering rate. Compared across inference-time collaboration, our WaltzRL training improves accuracy and reduces FTR, which leads to better efficacy as well as efficiency at inference-time. We are happy to provide further details if needed.
> >
> > > The experiments should be also benchmarked with other multi-agent RL frameworks, e.g., Zheng et al. 2024.
> >
> > As mentioned in L475, prior works such as Zheng et al., (2024) train multiple agents (e.g., an attacker and a defender) but only deploy one agent at inference (e.g., the trained defender). Moreover, their setting is formulated as a zero-sum game, whereas our work follows a positive-sum game. Due to these fundamental differences, we believe adapting these approaches into our setting, which deploys both agents to jointly enhance helpfulness and harmlessness, would require extensive modifications and may itself constitute interesting future work. We have already cited and discussed your mentioned prior works and will also add this discussion to Section 4 of our draft for further context.
> >
> > ---
> > We look forward to addressing any remaining questions during the rebuttal and hope these updates warrant a higher evaluation.

---

> ### Author Response · Authors · 2025-11-24
> **Follow-up to reviewer uDhH**
>
> Dear reviewer `uDhH`,
>
> Thanks again for your detailed feedback. We hope our responses addressed your concerns. Please let us know if you have any remaining questions or need further clarification, and we're happy to help!
>
> Best,
> Authors

---

> > ### Comment · Reviewer_uDhH · 2025-11-27
> >
> > Thank the authors for addressing my questions. I am satisfied with the comments and additional experiments. I will raise my score.

---

### Official Review · Reviewer_TwDJ · 2025-10-30

**Soundness:** 4
**Presentation:** 4
**Contribution:** 3
**Rating:** 6
**Confidence:** 5

**Summary:**

This paper introduces WALTZRL, a novel multi-agent reinforcement learning framework designed to address the fundamental tension between helpfulness and harmlessness in LLMs. The central problem is the dual challenge of models being susceptible to generating unsafe content while also exhibiting over-refusal on benign but sensitive prompts. WALTZRL reframes safety alignment as a collaborative, positive-sum game between a conversation agent, which generates responses, and a feedback agent, which provides constructive suggestions for improvement. Through extensive experiments across five diverse datasets, the authors demonstrate that WALTZRL significantly reduces both ASR and Over-Refuse Rat compared to strong baselines.

**Strengths:**

1. A key strength is the work's direct focus on reducing over-refusal. Unlike many safety methods that sacrifice helpfulness, this approach successfully improves the model's ability to respond to sensitive yet benign prompts.
2. The method is supported by comprehensive experiments across diverse benchmarks for safety, over-refusal, and general capabilities, showing clear improvements over strong baselines.

**Weaknesses:**

1. The experiments are limited to a single model (Llama-3.1-8B). This raises questions about how the approach generalizes to other model families or sizes. Including results on another model would significantly strengthen the paper's claims.
2. The paper only evaluates a single round of feedback (T_max=1), even though the framework supports more. The potential effects of multi-round interactions, including possible failure modes, are not explored.

**Questions:**

Did you try an ablation where you only train the feedback agent and keep the conversation agent frozen throughout? This would help clarify the benefits of co-evolving the agents versus simply pairing a static model with a trained feedback module.

---

> ### Author Response · Authors · 2025-11-20
> **Response to reviewer TwDJ**
>
> We greatly appreciate reviewer `TwDJ` for your detailed review. We are grateful that you find our work novel, with key strength being our work’s direct focus on reducing over-refusal, and our experiments extensive across five diverse datasets. We would like to clarify a few points below:
>
> > Did you try an ablation where you only train the feedback agent and keep the conversation agent frozen throughout? This would help clarify the benefits of co-evolving the agents versus simply pairing a static model with a trained feedback module.
>
> **TL;DR: we tested freezing the conversation throughout and found worse performance, showing the benefits of co-evolving agents.**
>
> Thank you for your insightful question. We have conducted the ablation where the conversation agent is frozen throughout the two stages of training and present the results below:
>
> | Method | WJ⬇️ | FH⬇️ | SR⬇️ | OB⬇️ | FB⬇️ | Avg ASR⬇️ | Avg ORR⬇️ | F1⬆️ |
> |---|---|---|---|---|---|---|---|---|
> | **Ours: WaltzRL** | **4.6** | **6.2** | **0.3** | **9.9** | **5.4** | **3.7** | **7.6** | **94.3** |
> | Frozen Conv. Agent | 8.1 | 6.6 | 0.6 | 22.4 | 6.0 | 5.1 | 14.2 | 90.1 |
>
> All numbers are in %. F1 is the harmonic mean of (1-average ASR) and (1-average ORR) and serves as an aggregate score of balancing helpfulness and safety, higher the better.
> As shown by the superior performance of WaltzRL across all 5 datasets compared to freezing the conversation agent throughout training co-evolving the agents via multi-agent joint training is crucial for the effectiveness, though only training the feedback agent still improves over inference-time collaboration (with an F1 of 87.9 vs 90.1 for frozen conversation agent).
>
> > The experiments are limited to a single model (Llama-3.1-8B).
>
> We appreciate your suggestion to evaluate on additional backbones. WaltzRL's multi-agent RL with multi-turn rollouts makes each training run computationally expensive, so we focused our experiments on Llama-3.1-8B given our compute constraints. However, we are actively seeking additional resources to conduct experiments on other model families and will update this thread if we have new results!
>
> We would also like to emphasize that **our framework is conceptually model-agnostic, and our approach does not involve any model-specific tricks**. We therefore expect WaltzRL to transfer well to other model families and sizes without modification. We will include this discussion in our manuscript, and, if accepted, we plan to try our best to add results on additional models in the camera-ready version. Thanks again for your suggestion.
>
> > The paper only evaluates a single round of feedback (T_max=1), even though the framework supports more.
>
> Thank you for raising this point. To clarify, **the goal of WaltzRL is to produce a safe and non-overrefusing response as quickly as possible** so that latency and token costs are minimized, instead of achieving this through a fixed number of feedback rounds. Therefore, fewer feedback rounds are always preferable, as they indicate more efficient convergence to the desired outcome.
>
> We set $T_{max}$ to 1 because: (1) As shown in Section 3, $T_{max}=1$ already performs extraordinarily well, outperforming strong baselines and achieving single-digits ASR and ORR. Thus we believe $T_{max}=1$ is sufficient in our current setting. (2) Additional interaction rounds increase inference cost at deployment. With adaptive feedback $T_{max}=1$, our inference cost is at most comparable to applying a safeguard model plus generating a revised response. In summary, **we experiment with $T_{max}=1$ because it’s both practically desirable and already empirically strong**. We will include this discussion in our draft for further context.
>
> ---
>
> We look forward to addressing any remaining questions during the rebuttal and hope these updates warrant a higher evaluation.

---

> > ### Comment · Reviewer_TwDJ · 2025-11-25
> >
> > Thank you for your detailed response and additional experiments. After consideration, I will keep my score.

---

### Official Review · Reviewer_WRqh · 2025-11-01

**Soundness:** 2
**Presentation:** 3
**Contribution:** 2
**Rating:** 4
**Confidence:** 3

**Summary:**

This work proposes a MARL framework to make the safety alignment as a positive-sum game by training a conversation agent and a feedback agent. The dynamic improvement reward encourages the conversation agent to take the feedback effectively. Experiments show results that unsafe and over-refusal responses are reduced for a better trade-off.

**Strengths:**

The paper is well-written and easy to follow. The novelty is sound, and the idea of game theory is interesting for the storytelling. The presentation of figures and algorithms are clear to me.

**Weaknesses:**

- Overall, I feel like the implementation is full of tricks and the details are not clear enough, which makes the story of game theory less convincing in practice.

- When gathering RL states and actions, it is expected to justify why only one random round is chosen as final feedback agent trajectory. Why not use the all T trajectories?

- Since the conversation agent trajectory is randomly chosen from A and B, more experiments need to show its benefits by comparing the cases where only A or B is taken.

- I feel like it is kind of what we want when initial responses from a conversation agent are already safe and not overrefusing, so the adaptive feedback needs clearer justification.

- In table 1, it seems the results of single-model RL and oracle label-converted feedback are better and might be more generalizable. More experiments of safety and over-refusal on different datasets are needed to show the generalizability.  Also the Pareto front is supposed to be a curve not a single point in the 2D plane for each method.

- Also, inference-time collaboration based on prompts seems to be more generalizable to me since it is training-free. It needs more justification regarding why RL post-training is needed.

- Figure 2 lacks confidence intervals or standard deviations, making it hard to distinguish between setups B and C.

**Questions:**

See Weakness

---

> ### Author Response · Authors · 2025-11-20
> **Response to reviewer WRqh (1/2)**
>
> We sincerely thank reviewer `WRqh` for your detailed feedback. We appreciate that you find our idea novel, the game-theoretic formulation interesting, and our draft well-written. As a reminder, our main contributions are proposing the WaltzRL framework to **jointly optimize two agents that co-evolves for safety**, the **novel DIR reward formulation**, and experiments demonstrating that WaltzRL **simultaneously reduces both unsafe responses and overrefusals against strong baselines**. We now address your concerns:
>
> > Overall, I feel like the implementation is full of tricks and the details are not clear enough, which makes the story of game theory less convincing in practice.
>
> Thank you for this feedback and we are happy to clarify each component of WaltzRL. Our core recipe consists of (1) the conversation-based rollout pipeline as the **collaboration protocol**; (2) the response reward and the Dynamic Improvement Reward design of the two agents to **encourage collaborative behavior**; (3) the adaptive stopping condition to **enhance practical efficiency**. While there are several components, we believe each component serves a well-defined objective. We will add explanations to Section 2 of our draft to clarify their motivations. If you would like further clarifications on any aspects, please let us know and we are happy to provide more details!
>
>
> > When gathering RL states and actions, it is expected to justify why only one random round is chosen as final feedback agent trajectory. Why not use the all T trajectories?
>
> Thank you for your question. We would like to first clarify that we sample one round as the feedback agent trajectory and sample one of $tau_A$ and $tau_B$ **to balance the number of trajectories for each agent** so that each agent has a single trajectory to train on for each global training step. Following the collaborative rollout stage, each agent is trained for a single policy gradient step, which keeps the samples on-policy. This is our current design and we have empirically shown in Section 3 that WaltzRL training is stable and results in effective models.
>
> As for the reason why not using all feedback rounds, because of the adaptive stopping condition, the number of rounds for each prompt will differ depending on whether the initial response requires improvement, creating trajectories with large length variance. **Directly training on all rounds would disproportionately upweight trajectories with multiple rounds, and potentially destabilize training**. Therefore, we sample one round of feedback for each trajectory to balance between longer and shorter trajectories. We will add these details to Section 2.3 of our draft to provide further context to readers.
>
> > Since the conversation agent trajectory is randomly chosen from A and B, more experiments need to show its benefits by comparing the cases where only A or B is taken.
>
> **TL;DR: We tested training with $\tau_A$ and $\tau_B$ only and found WaltzRL outperforms these settings.**
>
> Thank you for your suggestion. We have conducted further ablation experiments on training only using $\tau_A$ or $\tau_B$.
>
> | Method            | WJ⬇️     | FH⬇️     | SR⬇️     | OB⬇️     | FB⬇️     | Avg ASR⬇️ | Avg ORR⬇️ | F1⬆️      |
> |-------------------|---------|---------|---------|---------|---------|----------|----------|----------|
> | WaltzRL           | **4.6** | 6.2     | **0.3** | **9.9** | **5.4** | **3.7**  | **7.6**  | **94.3** |
> | Only use $\tau_A$ | 4.8     | **4.6** | 1.6     | 11.1    | 6.0     | 3.7      | 8.6      | 93.8     |
> | Only use $\tau_B$ | 8.6     | 8.6     | 0.3     | 12.7    | 5.8     | 5.8      | 9.2      | 92.4     |
>
> All numbers are in %. F1 is the harmonic mean of (1-average ASR) and (1-average ORR) and serves as an aggregate score of balancing helpfulness and safety, higher the better. Results show that ablating one of the two types of trajectories indeed achieves worse outcomes, illustrating the effectiveness of our mixed trajectory sampling strategy.

---

> ### Author Response · Authors · 2025-11-20
> **Response to reviewer WRqh (2/2)**
>
> > I feel like it is kind of what we want when initial responses from a conversation agent are already safe and not overrefusing, so the adaptive feedback needs clearer justification.
>
> Yes, you are exactly correct that it’s desirable to generate initial responses that are already safe and non-overrefusing. Note that the goal of our conversation-based protocol is not finding a safe and non-overrefusing response through fixed number of feedback rounds, but **producing a safe and non-overrefusing as soon as possible**, thus fewer feedback rounds are always more desired.
>
> In fact, our motivation for the adaptive feedback mechanism is that the **the initial conversation agent reward is already high at the end of training** (shown in Figure 4 Left in the appendix), indicating most responses do not need improvement and adaptive feedback can reduce latency and quality degradation.
>
> There will be two consequences if we omit the adaptive feedback mechanism. First, the feedback model will produce feedback to responses that are already satisfactory, incurring unnecessary cost. Furthermore, feedback will be activated for every prompt that is unrelated to safety, causing both latency and quality concerns on general prompts. As we show in Table 4, **our adaptive feedback mechanism is rarely triggered on general prompts unrelated to safety** (6.7% on AlpacaEval).
>
> Thanks again for your comment and we will include this discussion into Sections 2.4 and 3.3.
>
> > In table 1, it seems the results of single-model RL and oracle label-converted feedback are better and might be more generalizable. More experiments of safety and over-refusal on different datasets are needed to show the generalizability.
>
> Please note that Table 1 demonstrates **WaltzRL outperform both the single-model RL and oracle label-converted feedback baselines** on Attack Success Rate (ASR) and Over-Refuse Rate (ORR) across 5 diverse dataset (lower the better): on average WaltzRL achieves 3.7 ASR and 7.6 ORR, while single-model RL achieves 12.2 ASR and 8.6 ORR, and oracle label-converted feedback achieves 5.3 ASR and 14.9 ORR.
>
> We’d like to clarify that WaltzRL is only trained on the training set of WildJailbreak and OR-Bench, and we have already demonstrated in Table 1 that **WaltzRL generalize well to datasets that are out of distribution from training** (namely FORTRESS-Harm, StrongReject, and FORTRESS-Benign), outperforming strong baselines.
>
> > Also, inference-time collaboration based on prompts seems to be more generalizable to me since it is training-free. It needs more justification regarding why RL post-training is needed.
>
> We’d like to clarify that WaltzRL still deploys two agents at inference time (L87-90), and the inference-time collaboration baseline ablates the RL training part of our WaltzRL approach (L318-320). In Table 1, we have shown that **even on out-of-distribution datasets, WaltzRL still outperforms the inference-time collaboration baseline by a large margin**. This demonstrates the generalizability of WaltzRL. In Table 3 and Table 4, we show that the inference-time collaboration baseline w/o RL training has overly high feedback triggering rate and low label accuracy for feedback. It also frequently triggers feedback on non-safety related prompts in AlpacaEval and leads to quality degradations. Our WaltzRL approach fixes these issues through RL training.
>
> > Figure 2 lacks confidence intervals or standard deviations, making it hard to distinguish between setups B and C.
>
> Thank you for your suggestion. We would like to clarify the difference between Setups B and C are mainly on the label accuracy aspects: while setup B and C are indeed similar on %Improved and %Worsened metrics, the label accuracy remains consistently high in setup B while it degrades in setup C through training steps, with an about 20% accuracy gap on Figure 2 (right).
>
> ---
>
> We look forward to addressing any remaining questions during the rebuttal and hope these updates warrant a higher evaluation.

---

> ### Author Response · Authors · 2025-11-24
> **Follow-up to reviewer WRqh**
>
> Dear reviewer `WRqh`,
>
> Thank you again for your thoughtful feedback. We wanted to follow up on our responses and check if you have any remaining concerns or questions. We are happy to provide further clarifications as needed!
>
> Best,
> Authors

---

> > ### Comment · Reviewer_WRqh · 2025-11-26
> >
> > Thank you for your clarification and additional experiments. It seems my question regarding the Pareto front in weakness 5 is missing and not addressed: "Also the Pareto front is supposed to be a curve not a single point in the 2D plane for each method." Other than that, I do not have further concerns, and all the clarification should be reflected in the updated version.

---

### Meta-Review · Area_Chair_d6g3 · 2026-01-07

**Summary:**

This paper proposes a new multi-agent reinforcement learning framework, called WALTZRL, to balance the trade-off between helpfulness and harmlessness in LLMs, which is a challenging problem.

The reviewers appreciate the following strengths of the paper:

S1. The paper proposes a novel algorithm with several strengths, such as reducing over-refusal and effectively balancing helpfulness and harmlessness in LLMs.

S2. The paper provides extensive experiments across diverse benchmarks and conducts insightful analysis.

S3. The paper is easy to follow.

The ratings were mostly positive before the rebuttal. Reviewers uDhH and gQyF have explicitly indicated increases in their ratings to 6 and 8, respectively. Reviewer TwD provided a positive rating (6) before the rebuttal and is likely to raise or maintain their positive score. Reviewer p3hP is likely to raise their rating (from 4 to 6), since the concerns were clarified during the rebuttal.

In summary, the paper has overall positive ratings, with almost all raised concerns addressed, and the required changes can be readily incorporated into the camera-ready version. Thus, this paper is a candidate for final acceptance.

**Reviewer Concerns:**

The reviewers also raised the following concerns:

W1. Lack of ablation studies on trajectory sampling.

W2. Lack of experiments validating the necessity of co-evolving both agents.

W3. Lack of ablation studies of a variant using final-response reward instead of DIR.

The authors have successfully addressed most of these concerns during the rebuttal period by providing additional ablation studies on trajectory sampling, variants using final-response rewards, and the frozen conversation agent setting, and additional discussions and analysis.

**Reviewer Scores:**

The ratings were mostly positive before the rebuttal. Reviewer UaRs has explicitly indicated an increase in the rating to 8. Reviewers v3aX and WawF provided positive ratings (6, 6) before the rebuttal and are likely to raise or maintain their positive scores. Reviewer p3hP is likely to raise their rating (4 to 6) since the concerns were clarified during the rebuttal.

---

### Decision · Program_Chairs · 2026-01-26

Accept (Poster)